# Integrated Serosurveillance of Infectious Diseases Using Multiplex Bead Assays: A Systematic Review

**DOI:** 10.3390/tropicalmed10010019

**Published:** 2025-01-10

**Authors:** Selina Ward, Harriet L. S. Lawford, Benn Sartorius, Colleen L. Lau

**Affiliations:** UQ Centre for Clinical Research, Faculty of Health, Medicine and Behavioural Sciences, The University of Queensland, Brisbane, QLD 4006, Australia; h.lawford@uq.edu.au (H.L.S.L.); b.sartorius@uq.edu.au (B.S.); colleen.lau@uq.edu.au (C.L.L.)

**Keywords:** public health, malaria, neglected tropical diseases, vaccine-preventable diseases, vector-borne diseases, serological surveillance, IgG antibody seroprevalence

## Abstract

Integrated serological surveillance (serosurveillance) involves testing for antibodies to multiple pathogens (or species) simultaneously and can be achieved using multiplex bead assays (MBAs). This systematic review aims to describe pathogens studied using MBAs, the operational implementation of MBAs, and how the data generated were synthesised. In November and December 2023, four databases were searched for studies utilising MBAs for the integrated serosurveillance of infectious diseases. Two reviewers independently screened and extracted data regarding the study settings and population, methodology, seroprevalence results, and operational implementation elements. Overall, 4765 studies were identified; 47 were eligible for inclusion, of which 41% (*n* = 19) investigated multiple malaria species, and 14% performed concurrent surveillance of malaria in combination with other infectious diseases (*n* = 14). Additionally, 14 studies (29%) investigated a combination of multiple infectious diseases (other than malaria), and seven studies examined a combination of vaccine-preventable diseases. Haiti (*n* = 8) was the most studied country, followed by Ethiopia (*n* = 6), Bangladesh (*n* = 3), Kenya (*n* = 3), and Tanzania (*n* = 3). Only seven studies were found where integrated serosurveillance was the primary objective. The synthesis of data varied and included the investigation of age-specific seroprevalence (*n* = 25), risk factor analysis (*n* = 15), and spatial analysis of disease prevalence (*n* = 8). This review demonstrated that the use of MBAs for integrated surveillance of multiple pathogens is gaining traction; however, more research and capabilities in lower- and middle-income countries are needed to optimise and standardise sample collection, survey implementation, and the analysis and interpretation of results. Geographical and population seroprevalence data can enable targeted public health interventions, highlighting the potential and importance of integrated serological surveillance as a public health tool.

## 1. Introduction and Background

To monitor progress towards the prevention, control, elimination, and eradication of infectious diseases, accurate and timely surveillance is required [1]. To support the development of targeted public health interventions, population-level serological surveys (serosurveys) can provide rich data regarding immunity to infectious diseases from past infection and/or vaccination, and the current burden of chronic infections [2]. However, in low-resource settings, disease surveillance is challenged by limited time and resources, resulting in diseases being monitored in isolation [3,4]. Furthermore, as disease prevalence within populations decreases, it becomes increasingly difficult to identify remaining infections, due to more heterogenous and sparse patterns of disease [3]. One effective method to address these challenges is integrated serological surveillance (serosurveillance), whereby surveys test for multiple pathogens or species of pathogens using a single specimen [4]. Integrated serosurveillance offers many advantages compared to a siloed single-disease approach by saving both time and resources.

A commonly employed laboratory method for conducting serological analysis of samples for multiple pathogens simultaneously is through multiplex bead assays (MBAs), which utilise microscopic beads containing fluorescent dye and target molecules to allow for quantitative analysis of multiple targets [5]. For measuring immunoglobulin-G (IgG) antibodies, median fluorescence intensity (MFI) is considered proportional to levels of antigen-specific antibodies in the blood, and serves as a proxy for antibody titres [6]. This can be transformed quantitatively into seropositive and seronegative results and used to estimate specific pathogens at the population level [6]. This technique is less labour-intensive compared to traditional single-pathogen analysis such as ELISAs [7], and by analysing samples concurrently, measurement error and bias are minimised, since all samples are subjected to the same conditions [8].

The use of integrated serology-based epidemiology (sero-epidemiology) as a surveillance method can detect both high and low prevalence of infection [9]. When combined with geospatial and statistical modelling, seroprevalence data can also be used to produce important insights, such as identifying hotspots of disease occurrence or cold spots of low prevalence of immunity to vaccination-preventable diseases (VPDs). However, it is important to note that while this method of surveillance can be used as a complement to epidemiological monitoring, it cannot replace other methods such as the active or passive detection of clinical cases.

To the best of our knowledge, no previous reviews have examined integrated serosurveillance using MBAs. This review aimed to determine how MBAs have been used for integrated serosurveillance of infectious diseases and discuss the public health implications of utilising MBAs for integrated serosurveillance. Specifically, we aimed (i) to describe the pathogens studied using MBAs for serosurveillance, (ii) to describe the operational implementation of using MBAs for serosurveillance (sampling design, how the samples were collected, antibody targets, and laboratories where MBAs were conducted), and (iii) to describe the geographical distribution of studies and types of applications of MBAs for serosurveillance (synthesis of data, seroprevalence findings, and public health implications).

## 2. Methods

### 2.1. Protocol and Registration 

This systematic review’s protocol was registered with the International Platform of Registered Systematic Review and Meta-analysis Protocols (INPLASY) in April 2024 (INPLASY 202440094).

### 2.2. Literature Search

A systematic literature review was performed between November and December 2023 using the following databases: PubMed, MEDLINE, Scopus, Embase, and Cochrane Library. The full Boolean and search optimisation and refinement strategies used for each search can be found in Appendix A. Article screening and data extraction were completed using Covidence (Melbourne, Australia) [10] by two independent reviewers (SW and HL), and any discrepancies were resolved by a third reviewer (BS). This review was completed in accordance with the PRISMA checklist for systematic reviews [11], which is included in Appendix A.

### 2.3. Inclusion and Exclusion Criteria

All observational and intervention studies utilising MBAs for the concurrent serosurveillance of more than one infectious pathogen (or multiple species of a single pathogen) were included. The study inclusion criteria included the following: (i) the use of MBA technology, (ii) the examination of more than two infectious pathogens (or multiple species of a single pathogen), and (iii) the use of serological samples. There was no limit to the time of publication of studies. As the focus for this review was on the applied use of MBAs, we excluded studies that did not report any research data (commentaries, letters, editorials, viewpoints, reviews) and those with a limited focus (case reports, case series, outbreak reports). Studies not published in English or reporting indirect measures of seroprevalence (such as odds ratios for disease prevalence in certain areas compared to others) were excluded. Lastly, studies were excluded if they were conducted only for the validation or calibration of laboratory methods, without reporting prevalence.

### 2.4. Data Collection

Data were extracted using Covidence and collated into a tabular format to capture the following information: the author (year), study design, World Health Organization (WHO) region, country where the study was conducted, operational details of sample collection, integrated serosurveillance type (multiple species of one pathogen or multiple pathogens), diseases studied, antigens and/or antibodies used, collection of repeated samples over time, sampling design of the studies, participant demographics (target population), blood collection method, year of sample collection, year of sample testing, total number of participants (sample size), country/laboratory of sample analysis, disease prevalence findings, practical implications, and additional diagnostics used. A narrative summary of the data extracted from the studies is provided. Some studies examined pathogens other than neglected tropical diseases (NTDs), vaccine-preventable diseases (VPDs), and malaria, and were grouped as “other”. Studies were placed into categories based on the pathogens studied: (i) malaria (two or more species), (ii) VPDs (two or more pathogens), (iii) a combination of NTDs/VPDs/Other, and (iv) malaria plus the combination of NTDs/VPDs/Other.

### 2.5. Risk of Bias and Quality Assessment

A quality assessment and a risk of bias assessment were conducted using the risk of bias in systematic reviews (ROBIS) tool (Appendix A) [12]. As the aim of this review was to examine how MBAs have been used for the integrated serosurveillance of infectious diseases, a meta-analysis was not conducted. Additionally, due to the high heterogeneity of papers (survey design, pathogens examined, location) a meta-analysis was not considered feasible. Furthermore, an individual study assessment of risk of bias would not impact the results.

## 3. Results

A total of 4765 records were identified, of which 956 were automatically detected as duplicates through the Covidence application (Melbourne, Australia) [10]. Following title and abstract screening, 354 remained for full text screening. Finally, 47 studies fulfilled the inclusion criteria (Figure 1) [13,14,15,16,17,18,19,20,21,22,23,24,25,26,27,28,29,30,31,32,33,34,35,36,37,38,39,40,41,42,43,44,45,46,47,48,49,50,51,52,53,54,55,56,57,58,59].

### 3.1. Location of Studies and Time from Sample Collection to Publication

Studies were identified from 30 countries. When classified into regions, twenty studies (66%) were conducted in the African region, twelve in the Americas region (25%), eight in the Western Pacific region (17%), four in the European region (8%), and three in the South-East Asian region (6%). Haiti (*n* = 8) was the most studied country, followed by Ethiopia (*n* = 6), Bangladesh (*n* = 3), Kenya (*n* = 3), and Tanzania (*n* = 3). The median time between the collection of samples and publication of studies was four years (range: 1–14 years). The earliest study was published in 2011, with peak number of studies (*n* = 13) published in 2022, decreasing to only 5 studies in 2023 (Figure 2). This shows an increasing trend in publications. Most studies (*n* = 19) focused on malaria (two or more species), followed by NTDs/VPDs/Other (*n* = 14), and malaria plus the combination of NTDs/VPDs/Other (*n* = 7). Seven studies focused on VPDs (two or more pathogens).

### 3.2. Pathogens and Antigens Examined

A full list of the pathogens examined and the antigens used in each study can be found in Appendix A. Overall, the most studied pathogen was *Plasmodium falciparum* (*n* = 26), followed by *Plasmodium vivax* (*n* = 19) and *Plasmodium malariae* (*n* = 15). Regarding NTDs, the most studied were *Wuchereria bancrofti* (*n* = 9), *Brugia malayi* (*n* = 9), *Strongyloides stercoralis* (*n* = 7), and *Chlamydia trachomatis* (*n* = 6). *Cryptosporidium parvum* (*n* = 7), enterotoxigenic *Escherichia coli* (*n* = 5), *Entamoeba histolytica* (*n* = 5), and *Toxoplasma gondii* (*n* = 5) were the most studied within the ‘other’ category.

### 3.3. Sampling Methods

One study included females of child-bearing age only, while all other studies included both genders. Approximately one third of studies targeted only children under 18 years (*n* = 15; 31%). One study did not define the target population of the samples used within the study. Further details regarding sampling methods and the rationale behind this are detailed in Table 1. For 16 studies, the target pathogen for the primary study was the same as the pathogen(s) examined using MBAs; however, this was mostly utilised by studies examining malaria (*n* = 13) [13,19,20,31,32,34,36,37,39,40,44,49,56], LF [22,57], and VPDs [17].

### 3.4. How Samples Were Collected

Only seven studies (15%) were undertaken where integrated surveillance was the primary objective of the study [17,18,29,31,32,36,41], rather than a secondary analysis of data from another study (Table 1). Most studies were performed using samples collected from a previously published study (*n* = 25; 54%). Many studies were also performed using samples collected from public health surveys for lymphatic filariasis (LF) (*n* = 3) [22,48,57], malaria (*n* = 4) [13,24,37,40], HIV/AIDS (*n* = 2) [53,60], trachoma (*n* = 1) [28], and onchocerciasis (*n* = 1) [27], or undertaken in conjunction with an emergency nutrition assessment survey (*n* = 3) [25,26,38]. Regarding the study setting, most were household-based recruitment (*n* = 23), followed by clinic-based (*n* = 8) and school-based recruitment (*n* = 6), or other (or not defined) (*n* = 10). Most studies collected samples using dried blood spots (DBSs) (*n* = 29; 61%) or serum samples (*n* = 14; 29%) [21,29,32,33,34,35,41,42,45,46,49,52,57,58]. Some studies used both DBSs and serum samples (*n* = 3) [20,44,53], and one study did not define the sample collection method [17].

### 3.5. Determining Cut-Offs for Seropositivity and MBA Antigens Used

Multiple methods were used to determine seropositivity (Appendix A). The most common method was the presumed unexposed method (*n* = 28), followed by mixture models (*n* = 13), receiver operating characteristics (ROC) curves (*n* = 10), and standard units (*n* = 8). Note that some studies used multiple methods for determining seropositivity; thus, the total exceeds 47. Under the presumed unexposed method, seropositivity cut-off values are generally calculated based on MFI levels measured from non-endemic regions of the pathogen of interest [6], whereas mixture models are a form of probabilistic statistical modelling that assume the presence of two or more distinct groups, generally seronegative and seropositive within the sample population [6].

### 3.6. Laboratories Where MBAs Occurred

Most samples were analysed at the US Centers for Disease Prevention and Control (CDC) (*n* = 29; 61%). Other laboratories included the Kenya Medical Research Institute in Nairobi [29,47], the CDC in Nigeria [30], the University of Philippines [39], the University of São Paulo, Brazil [44], the National Laboratory of Haiti [56], Institut Pasteur in France [58], the Scientific Institute of Belgium [17,21], the London School of Hygiene and Tropical Medicine [19,20], and the University of Oxford in the UK [41].

**Table 1 tropicalmed-10-00019-t001:** Summary of studies included.

Author, year, [ref] WHO Region (Country)	Operational details of sample collection with sampling method/s (references for information extraced using a secondary reference)
Malaria (two or more species)	
Assefa 2019 [13] Africa (Ethiopia)	Malaria Indicator Survey (2015)—A nationally representative cross-sectional survey using a two-stage cluster approach for the selection of enumeration areas and random sampling for household selection targeting individuals of all ages [61].
Byrne 2022 [19] Western Pacific (Lao)	A cross-sectional household (conducted in 2016) survey (aged 18+ months) targeting four districts in Northern Lao with known malaria hotspots. A stratified two-stage cluster sampling design chose 25 survey clusters based on proportional-to-population size; then, households were selected randomly [62].
Byrne 2023 [20] Western Pacific (Malaysian Borneo)	An environmentally stratified, population-based cross-sectional study (conducted in 2015) aiming to understand the transmission of malaria in people aged over 3 years, in northern Sabah, as part of the MONKEYBAR project. A non-self-weighting two-stage sampling design was employed whereby villages were selected based on the proportion of forest cover, and 20 households within each village were randomly selected [63].
Feleke 2019 [27] Africa (Ethiopia)	Convenience sampling of individuals (5+ years) within a community-based onchocerciasis survey in three villages (Arengama 1, Arangama 2 and Konche) of southwest Addis Ababa. Participants were selected through convenience sampling in 2016.
Herman 2023 [30] Africa (Nigeria)	Nigeria HIV/AIDS Indicator and Impact Survey (NAIIS)—Conducted in 2018, this was a nationally representative cross-sectional survey using a two-stage cluster approach for the selection of enumeration areas and random sampling for household selection targeting individuals of all ages [64].
Jeang 2023 [31] Africa (Ethiopia)	**Primary study:** A cross-sectional serosurvey conducted in 2018 in two areas of southwestern Ethiopia (Arjo-Didessa and Gambella) with contrasting malaria transmission intensities. The study usedhousehold clustering, and individuals aged over 15 years were invited to participate.
Khaireh 2012 [32] Africa (Republic of Djibouti)	**Primary study:** A cross-sectional household-based serosurvey conducted in 2002 investigating the prevalence of malaria infections in adults aged 15–54 years in the Republic of Djibouti. Cluster sampling was used to select households whereby anonymized samples were collected from adults aged 15–54 years.
Koffi 2017 [34] Africa (Cote d’Ivoire)	In 2010, 2012, and 2013, patients presenting to Abobo healthcare facility with symptomatic malaria were invited to join the study. In 2011, due to the Civil War, additional young asymptomatic school children (6–15 years) were recruited in a transversal survey (*n* = 207) [65].
Labadie-Bracho 2020 [36] Americas (Suriname)	**Primary study:** A cross-sectional study of adults (>18 years) conducted in three regions of Suriname (Stoelmanseiland and surrounding areas (Gakaba, Apoema, and Jamaica), Benzdorp) between 2017 and 2018. Stoelmanseiland: All villagers who could be reached within a 4-day enrolment period were eligible to participate. Benzdorp: Participants were recruited during active case detection conducted by the National Malaria Programme.
Leonard 2022 [37] Africa (Ethiopia)	Malaria Indicator Survey (2015)—A nationally representative cross-sectional survey using a two-stage cluster approach for the selection of enumeration areas and random sampling for household selection targeting individuals of all ages [61].
Lu 2020 [38] South-East Asia (Bangladesh)	Conducted in 2018, this vaccine coverage survey and vaccine serosurvey was conducted in two areas of Cox’s Bazar, Bangladesh (Nayapara and makeshift settlement camps (MSs)) alongside an Emergency Nutrition Assessment. This assessment used the Standardized Monitoring and Assessment for Relief and Transitions (SMART) Methodology. In Nayapara, households were selected using simple random sampling whereby in MSs, a cluster sampling design was used [66].
Macalinao 2023 [39] Western Pacific (Philippines)	A rolling cross-sectional survey was conducted in three municipalities in three provinces in the Philippines. Sites were selected based on the 2014 Philippines’ National Malaria Program operational definition of malaria-endemic provinces (high endemicity—Rizal, Palawan; sporadic local cases—Abra de Ilog, Occidental Mindoro; malaria-free—Morong, Bataan). Participants were recruited from healthcare facilities between 2016 and 2018 [67].
McCaffery 2022 [40] Africa (Ethiopia and Costa Rica)	Ethiopia: Ethiopia Malaria Indicator Survey 2015 [61] Costa Rica: A household-based serosurvey conducted 2015 in the Costa Rican canton of Matina (2015). This location was selected as one of the last locations in the country where malaria cases were present. Ethiopia (*n* = 7077) was selected, as the country is co-endemic for both P. falciparum and P. vivax, and Costa Rica (*n* = 851) was selected as a representative low-endemic region for mono-species Plasmodium.
Monteiro 2021 [44] Americas (Brazil)	Biorepository at the Institute of Biomedical Sciences of the University of São Paulo containing samples from three studies: Curado et al. (1997): A cross-sectional (1992–1994) household-based study of adults (15+ years) within an 5–10 km area of a malaria case in two areas in Brazil. Silva-Nunes et al. (2006): A cross-sectional, household-based study describing baseline malaria in rural Amazonia in 2004. Medeiros et al. (2013): A cross-sectional study examining symptomatic and asymptomatic malaria (aged <1 year) cases in four localities of the Madeira River [68,69,70].
Oviedo 2022 [48] Americas (Haiti)	Transmission assessment survey (TAS) for LF: Conducted yearly (target population 6–7 years) between 2014 and 2016, implementation units were chosen if they had completed at least five rounds of annual MDAs with a coverage rate ≥ 65% of the entire population of the implementation unit [71,72].
Perraut 2017 [49] Africa (Senegal)	Yearly cross-sectional sampling (between 2002 and 2013) of febrile individuals (all ages) presenting to free medical clinics in two rural villages (Dielmo and Ndiop) in Senegal [73,74].
Rogier 2017 [54] Africa (Mali)	The primary study was a matched-control trial to assess the effectiveness of a school-based WASH programme (2014) in southern Mali. Serology samples were collected from school children (4—17 years) as part of a cross-sectional study nestled within longitudinal impact evaluation (NCT01787058) [75].
Rogier 2019 [53] Americas (Haiti)	A 2017 serosurvey (ages > 6 years) for malaria conducted in Haiti. The sampling method was not defined.
Van den Hoogen 2021 [56] Americas (Haiti)	Based in Grand’Anse, in the southwest of Haiti, a prospective healthcare facility-based case–control study was conducted in 2018. Cases (>6 months) were defined as a febrile individual with positive RDT results for malaria. Controls were RDT-negative individuals matched via age and gender [76].
VPDs (two or more pathogens)	
Boey 2021 [17] Europe (Belgium)	**Primary study:** A cross-sectional study (coducted 2014–2016) examining vaccination and immunity to vaccine-preventable diseases in adult (>18 years) at-risk patients attending outpatient clinics at the University Hospitals of Leuven.
Breakwell 2020 [18] Western Pacific (Solomon Islands)	**Primary study:** Conducted in 2016, a national cross-sectional school-based cluster survey of children (6–7 years) in the Solomon Islands. Schools (*n* = 80) were systematically chosen proportional to the estimated population size, and all eligible children were invited to participate.
Cabora 2016 [21] Europe (Belgium)	Randomly selected leftover serum samples (*n* = 670) collected in 2012 by six laboratories in three regions of Belgium (Flanders, Wallonia, and Brussels Capital Region). Samples (*n* = 1500) were collected from “healthy” asymptomatic (for pertussis) adults aged 20–30 years [77].
Feldstein 2020 [26] South-East Asia (Bangladesh)	Emergency Nutrition Assessment (2018)—This survey used data and samples from the study described by Lu et al. (2020). In brief, a survey was conducted in two areas of Cox’s Bazar, Bangladesh (Nayapara and makeshift settlement camps (MSs)) alongside an Emergency Nutrition Assessment [66].
Khetsuriani 2022 [33] Europe (Ukraine)	Conducted in 2017, this cross-sectional survey investigated immunity (children aged 2–10 years) to polioviruses following an outbreak of cVDPV1 in four areas (Zakarpattya province, Sumy province, Odessa province, and Kyiv City) with one-stage cluster sampling in provinces and secondary stratified simple random sampling [78,79].
Minta 2020 [43] Americas (Haiti)	A nationally representative household-based survey (conducted in 2017) designed to estimate chronic hepatitis B virus infection and immunity to diphtheria, tetanus, measles, and rubella in children aged 5–7 years. Sampling was performed using a two-stage cluster sampling method whereby enumeration areas were selected proportional to size and households were selected at random [80].
Rogier 2019 [53] Africa (Nigeria)	2018 Nigeria HIV/AIDS Indicator and Impact Survey—A nationally representative cross-sectional survey using a two-stage cluster approach for the selection of enumeration areas and random sampling for household selection targeting individuals of all ages [64].
Combination of NTDs/VPDs/Other	
Aiemjoy 2020 [14] Africa (Ethiopia)	A cluster-randomised trial (conducted in 2015) designed to determine the effectiveness of a comprehensive WASH package for ocular *C. trachomatis* infection (NEI U10 EY016214) in children (0–9 years) in Amhara, Ethiopia. A random sample of 60 children in each cluster were chosen for inclusion [81].
Arnold 2019 [15] Africa (Haiti, Tanzania, and Kenya)	Samples from previous studies. Haiti: A household-based cohort study examining LF prevalence in young children (0–4 years) at 6–9-month intervals in the coastal town of Leogane, Haiti, between 1990 and 1999. Households were selected based on previous high prevalence of LF (1990 and 1991). Kenya: A randomly selected, household-based randomised controlled intervention trial to assess the impact of ceramic water filters on the prevention of diarrhoea and cryptosporidiosis in young children (4–10 months) in Siaya county of western Kenya. Tanzania: A household-based, randomized controlled trial of children (1–9 years) examining the effects of annual azithromycin distribution (2012–2015) on trachoma in 96 independent clusters from the Kongwa district of Tanzania [82,83,84].
Cadavid Restrepo 2022 [22] Western Pacific (American Samoa)	Transmission assessment survey (TAS) for LF: Data were obtained from three TASs conducted across elementary schools (children aged 6–7 years) in American Samoa (all schools on the main island of Tutuila and the adjacent island of Aunu’u) in 2011, 2015, and 2016. Surveys were carried out at 25, 30, and 29 schools for each survey year, respectively [71].
Chan 2022 [23] Western Pacific (Malaysia)	An environmentally stratified, population-based cross-sectional study (2015) aiming to understand the transmission of malaria in people aged over 3 years, in northern Sabah, as part of the MONKEYBAR project. A non-self-weighting two-stage sampling design was employed whereby villages were selected based on the proportion of forest cover, and 20 household within each village were randomly selected [63].
Cooley 2021 [25] South-East Asia (Bangladesh)	Emergency Nutrition Assessment (2018)—This survey used data and samples from the study described by Lu et al. (2020) [66].
Fornace 2022 [28] Africa (Ghana)	Trachoma pre-validation: A population-based survey (targeting children 1–9 years) conducted in the Northern, Northeast, Savanna, and Upper West regions of Ghana between 2015 and 2016 using a two-stage cluster sampling approach. Villages were chosen proportional to population size, and households were selected using compact segment sampling [85,86].
Fujii 2014 [29] Africa (Kenya)	**Primary study (2011):** Random selection (based on site, sex, and age group) of participants (all ages) based on the Health and Demographic Surveillance System database of two Kenyan towns, Kwale and Mbita.
Kwan 2018 [35] Africa (Tanzania)	The samples were leftover serum s from the Mother–Offspring Malaria Study Project, a large cohort study conducted from 2002 to 2006 in Muheza, northeastern Tanzania, an area of intense malaria transmission. Pregnant mothers (18–45 years) were recruited pre-natally at the Muheza Designated District Hospital [87].
Mentzer 2022 [41] Europe (United Kingdom)	**Primary study:** The UK Biobank is a large prospective study with over 500,000 participants aged 40–69 years, recruited between 2006 and 2010. Participants were recruited from 22 assessment centres throughout the UK, covering a variety of different settings to provide socioeconomic and ethnic heterogeneity and an urban–rural mix. [88]
Miernyk 2019 [42] Americas (Alaska)	A cross-sectional survey investigating exposure to highly pathogenic avian influenza virus H5N1 in individuals and their families (>5 years) in Anchorage and western Alaska between 2007 and 2008. Participants were grouped into four groups that had regular contact with wild birds: (i) rural subsistence bird hunters and (ii) their family members, (iii) urban sport hunters, and (iv) wildlife biologists [89].
Mosites 2018 [45] Americas (Alaska)	This survey used data and samples from the study described by Miernyk et al. (2019) [89].
Won 2018 [57] Western Pacific (American Samoa)	Transmission assessment survey (TAS) for LF: Data were obtained from 3 TASs conducted across elementary schools (children aged 6–7 years) in American Samoa (all schools on the main island of Tutuila and the adjacent island of Aunu’u) in 2011 and 2015. The target sample sizes in 2011 and 2015 were 1042 and 1014, respectively [71].
Woudenberg 2021 [58] Europe (France)	A mix of samples from the routine hospital medical care (2002–2020) (94.4%) of individuals of all ages (0–100 years) and samples collected in the INCOVPED (NCT04336761) study; an observational study examining COVID-19 prevalence in children presenting to emergency departments in north-eastern France (COVID-19 and seasonal Human Coronaviruses).
Zambrano 2017 [59] Africa (Rwanda)	A nested village-level study (10 households with at least one child <4 years) within a cluster-randomised trial (2014–2016) examining improved cookstoves and household water filters. The larger trial consisted of 96 sectors in western region of Rwanda. Selection for inclusion within the nested-study was determined using a stratified, two-stage design—purposive sampling of study areas, then random sampling of households [90].
Malaria plus combination of NTDs/VPDs/Other	
Arzika 2022 [16] Africa (Niger, Malawi, and Tanzania)	A pre-specified, secondary analysis of the MORDOR Niger trial (CT02048007) where 30 rural communities were randomised 1:1 to biannual mass azithromycin distribution or a placebo offered to all children aged 1–59 months (0–4 years). A sub-study examining morbidity randomly selected 30 communities from the trial, and from this, 50 children within each community were asked to provide samples. This study was conducted between 2014 and 2020 [91,92].
Chan 2022 [24] Americas (Haiti)	Under the Global Fund grant against malaria, enumeration areas (117 communities) throughout Haiti were selected based on high risk of malaria. Households (20 per enumeration area) were chosen at random, and people of all ages were invited to participate. This study was conducted between 2014 and 2015.
Moss 2011 [46] Americas (Haiti)	A randomised, placebo-controlled trial conducted between 1998 and 1999 investigating the tolerance, efficacy, and benefit of combining the chemotherapeutic treatment of intestinal helminths and lymphatic filariasis in the town of Leogane, Haiti, in 12 selected primary schools (children aged 5–11 years) [93].
Njenga 2020 [47] Africa (Kenya)	A cross-sectional study (2015) to conduct an epidemiological assessment of LF infection in people aged over 2 years before restarting MDAs. Ten sentinel sites in costal Kenya were selected and five were selected based on LF risk. Household sampling was conducted to achieve a sample size of 300/sentinel site [71,94].
Plucinski 2018 [50] Africa (Mozambique)	**Primary study:** Two consecutive cross-sectional household surveys (2013 and 2014) before and after the LLIN campaign in six rural districts of the northern province of Nampula. From each district, 20 survey clusters were chosen randomly, with random selection of households within each cluster.
Poirier 2016 [51] Americas (Haiti)	A household-based longitudinal cohort study (*n* = 61) based in the coastal town of Ca Ira examining the effect of diethylcarbamazine (DEC)-fortified salt on the transmission of LF in children (2–10 years) at time points of 2011, 2013, and 2014. Additional samples (*n* = 127) were collected from the above cohort in 2014, as were samples from additional children (2–10 years) in Ca Ira.
Priest 2016 [52] Western Pacific (Cambodia)	A nationally representative survey (conducted in 2012) designed to estimate serological evidence in vaccine-preventable diseases in women of childbearing age (15–39 years). Based on the 2009 Cambodian neonatal tetanus risk assessment, multi-stage cluster sampling was performed with oversampling of areas with higher risk of tetanus [95,96].

### 3.7. Synthesis of Data

Several methods of data synthesis were performed, including age-specific seroprevalence (*n* = 17) or risk factor analysis (*n* = 10). Seven studies estimated seroprevalence by geographic region, two studies provided maps of seroprevalence, and nine studies used spatial analysis methods, of which one was for the identification of hotspots. Four studies were used to assess the impact of an intervention. Five studies described the correlation of antibody seropositivity with another diagnostic method (polymerase chain reaction (PCR), parasitaemia, and vaccine history). Figure 3 summarises the data synthesis methods used, and further information can be found in Appendix A.

### 3.8. Prevalence Findings

Most studies reported antigen seroprevalence (Appendix A). Some studies focused on high prevalence of disease, often in known endemic areas, the most common example being malaria [31,36,46,97,98], while others focused on low prevalence of diseases that have been targeted for elimination, such as yaws or trachoma [25,28]. An example of this is the low serological prevalence of yaws among refugee populations in Cox Bazar, where seropositivity for both *Rp17* and *TmpA* antigens was found to be 0% (95% CI: 0–1.7) in Nayapara and 0% (95% CI: 0–1.1) in makeshift settlements [25].

### 3.9. Potential Public Implications

Two studies used the data generated to provide national-level estimates of disease seroprevalence, while two other studies used the data to provide vaccination coverage estimates. Most study findings have public health implications related to age-specific immunity or seroprevalence (*n* = 27), location-specific immunity or seroprevalence (*n* = 26), or data regarding risk factor analysis (*n* = 16) (Figure 4). Among studies evaluating public health interventions or programmes (*n* = 5), two focused on the efficacy of Water, Sanitation, and Hygiene (WASH) programmes [14,54], two on the effectiveness of water filters [15,59], and one on the impact of long-lasting insecticide-treated bed nets [50].

## 4. Discussion

Our review showed that MBAs have mostly been used for the integrated serosurveillance of malaria with less emphasis on other infectious pathogens. Most studies use data from existing or previously published studies rather than standalone integrated serosurveys. Data synthesis ranges from estimating age-specific seroprevalence and risk factor analysis to spatial analysis and/or hotspot identification. Given the growing utilisation of MBAs for serosurveillance, further research is needed to optimise operational components and laboratory analysis. Additionally, expanding the network of laboratories capable of conducting MBAs is crucial, as most studies to date have relied on laboratory analyses conducted at the US CDC.

This review found a plethora of antigens used to examine the seroprevalence of malaria that targets different species (e.g., species-specific *MSP1-19*), and different life stages of the parasite (e.g., *AMA* targeting the schizont stage, or *CSP* targeting the sporozoite stage). This may potentially be due to the greater morbidity and mortality burden associated with malaria and the abundant funding support for malaria research compared to other infectious pathogens [99,100,101]. In 2019 alone, there were an estimated 608,000 deaths caused by malaria globally [102], compared to 36,055 deaths from dengue [103] and 35,000 from tetanus [104]. This relates back to the basics of disease surveillance, as stated by Foege, Hogan, and Newton in 1976: “the reason for collecting, analysing, and disseminating information on a disease is to control that disease”. To extrapolate, infectious pathogens that are harder to control, such as dengue [105], may be less prioritised than other infectious pathogens that have the potential to be controlled through readily expendable public health programmes such as chemoprophylaxis and bed-net distribution programmes.

For VPDs, the most common form of data synthesis was the estimation of seroprevalence by age and region to determine immunisation coverage and herd immunity. The use of serology for monitoring the population dynamics of VPDs has been shown to provide information regarding the age- and population-group-specific waning of immunity [4,106], which is especially important when considering the risk of future outbreaks. Furthermore, many studies have shown that serological surveillance for vaccine coverage provides higher accuracy than vaccination records [107], especially in low- and middle-income countries (LMICs) [108] where data may be limited, incomplete or hard to access. Information regarding population-level serological immunity is useful, in combination with other vaccination data, for influencing and changing vaccination schedules or promoting vaccination top-up campaigns to areas where this is most needed.

The WHO Toolkit for Integrated Serosurveillance of Communicable Diseases in the Americas [109] has emphasised the importance of integrated serosurveillance; however, in our review, only seven studies were conducted in which integrated serosurveillance was the primary objective. While not explicit, this has the potential to introduce sampling bias, for example, if assessing a public health intervention that was targeted to higher-risk individuals for the disease(s) of focus. Comparatively, for studies using a single-disease-focus sampling design that is extrapolated to multiple pathogens, there is much greater risk of introducing bias through targeted sampling methods that may potentially lead to misestimation of true disease prevalence [109]. For example, if using samples acquired from a previously conducted study examining measles, the target population will likely be children aged under 5 years. If these samples are then utilised for secondary integrated serological surveillance examining additional VPDs such as influenza or NTDs such as LF, there may be underestimation due to age-related the differences. Additionally, a recent systematic review examining bias in population-level measles serosurveys found that the majority of studies introduced moderate-level sampling bias by utilising a restricted, non-representative sample, such as convenience sampling [110]. While this may not be the case for the studies included in this review, it does highlight important considerations when using samples from repositories or biobanks to make more generalised population-level inferences. Nonetheless, the advantages of incorporating serosurveillance into routine surveys, such as health and demographic surveys, likely outweigh the disadvantages that may be introduced through bias.

Another important operational consideration is the physical method of sample collection. Obtaining venous blood samples requires training in phlebotomy, often by nurses, doctors, or laboratory technicians, which may be limited in rural and remote settings or in LMICs. In contrast, many of the samples examined within this review were DBSs prepared using fingerprick samples that can be collected by individuals without formal training. For MBAs, the use of DBSs has been compared with the use of serum samples, and DBSs showed comparable high sensitivity and specificity [111]. For example, a study comparing DBSs to serum samples for SARS-CoV-2 testing using MBAs found a specificity of 99.5% for both the N and S1 antigens [112]. In the same study, specificity ranged from 83% for samples collected 0–20 days post-symptomatic infection onset to 95% for samples collected 61–90 days post-infection onset [112]. Another example of this pertains to a study examining the use of MBA testing precision between DBSs and serum samples for HIV-1, which found that DBS antibody correlation ranged from 0.87 to 0.98 and serum samples raged between 0.90 and 0.97 [113]. In addition to the ease of collection and performance of specimens, when kept under ideal conditions between −20 °C and −80 °C, DBSs have the potential to remain viable for up to 20 years [114].

It is important that national capacity is improved to enable sample analysis at a regional or country level to avoid bottlenecks in data analysis and result dissemination. While most studies utilised laboratory facilities at the US CDC, it is encouraging that some studies undertook sample analysis at in-country laboratories. This aligns well with the WHO toolkit for integrated serological surveillance [109], and the 2030 NTD elimination framework [115,116,117] which, as outlined in the Ending the neglect to attain the Sustainable Development Goals: A road map for neglected tropical diseases 2021–2030, aims for 90% of endemic countries to utilise an integrated approach for NTD surveillance and integrated control by the year 2030 [117]. For this to be achieved, high levels of cross-collaboration and donor support are required to facilitate the upskilling and capacity building of local researchers and clinicians. However, moving forward, it would be beneficial to increase the capacity for laboratory analysis to be undertaken in more locations, within or closer to the countries in which the studies are being conducted. Furthermore, our study found that the median time between sample collection and publication was four years. By increasing the regional and national capacity for sample collection and analysis, this could decrease the time taken for the results to be available and for findings to be used for public health decision making. This is particularly important for infectious diseases near elimination- or outbreak-prone diseases that require a timely response. Furthermore, it is important to ensure that suitable antigens are widely available for relevant infectious pathogens to enable use in surveys in all areas that may be affected.

This systematic review has several limitations. Firstly, to expand the scope of this search, we included studies that examined different species of a single pathogen, whereas integrated serosurveillance is often considered for the concurrent serosurveillance of many different pathogens. However, this review has demonstrated that as time has progressed, the number and range of pathogens studied has steadily increased. This also highlights the potential publication bias, as the large range of time between sample collection and study publication may have impacted the number of studies retrieved within this review. Another limitation of this study is the inconsistent reporting of sample collection within each study. While many clearly detailed the process by which samples were collected, others referred to previously published studies.

## 5. Conclusions

In conclusion, this review has shown that MBA usage for the integrated serosurveillance of pathogens is gaining traction; however, most studies used MBAs to estimate the seroprevalence of malaria and used data from existing or previously published studies. Ensuring antigens are widely available can enable the implementation of primary serosurveillance studies of relevant infectious pathogens of importance. This can be further supported by expanding the network of laboratories capable of conducting MBAs. It is important to note that integrated serosurveillance can be used as a complement to epidemiological monitoring, but cannot replace other methods such as the active or passive detection of clinical cases. Nonetheless, the benefits of integrated serosurveillance as a public health tool are clear; a deeper understanding of geographical variation in prevalence and associated contextually relevant risk factors for infection can help identify populations eligible for future vaccination and targeted public health interventions.

## Figures and Tables

**Figure 1 tropicalmed-10-00019-f001:**
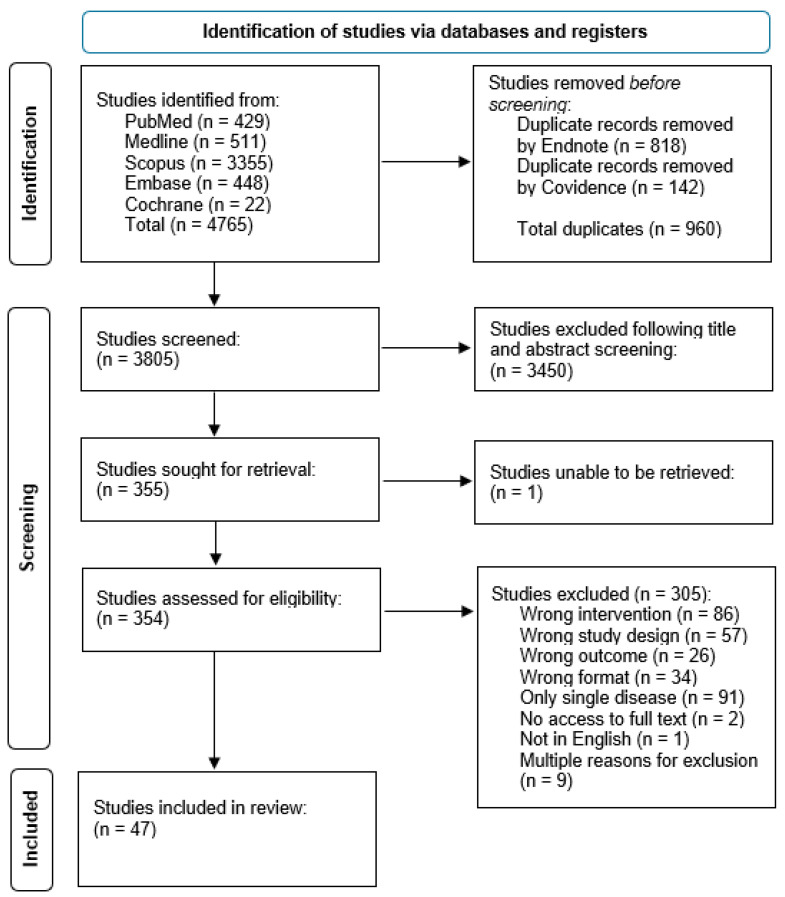
PRISMA flow diagram of studies included in this review.

**Figure 2 tropicalmed-10-00019-f002:**
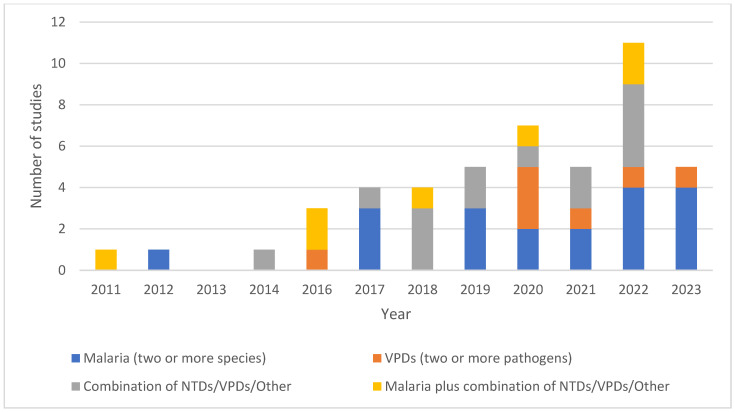
Number of included studies published each year by disease category: malaria (two or more species); VPDs (two or more pathogens); combination of NTDs/VPDs/Other; malaria plus combination of NTDs/VPDs/Other.

**Figure 3 tropicalmed-10-00019-f003:**
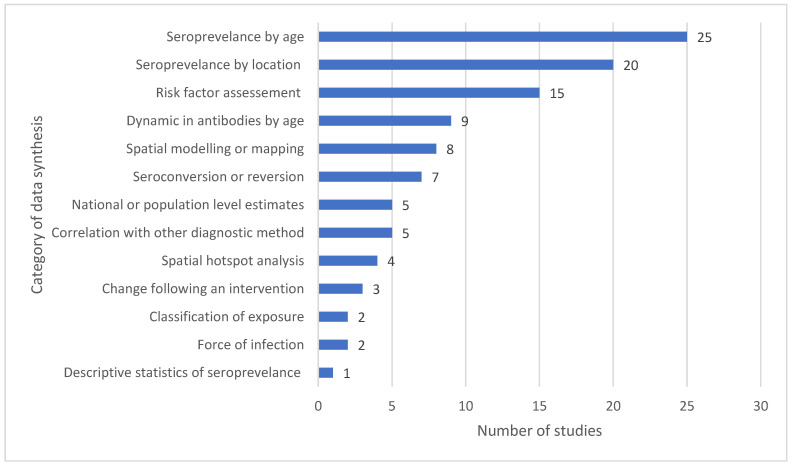
Categories of the data synthesis of the included studies. Some studies belong to more than one category; thus, the total exceeds the number of included studies.

**Figure 4 tropicalmed-10-00019-f004:**
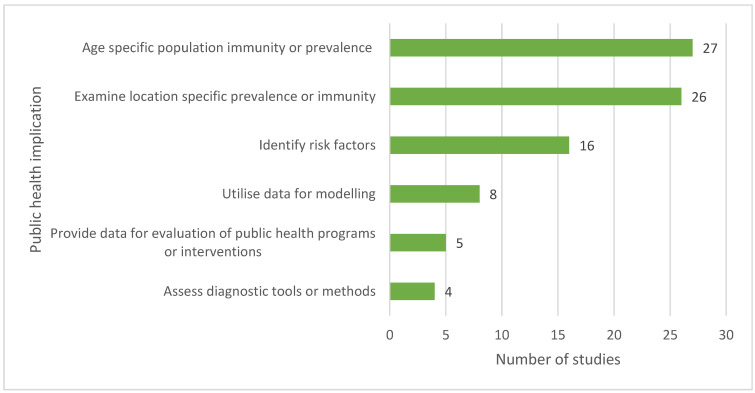
Potential or reported public health implications of the included studies. Some studies belong to more than one category; thus, the total exceeds the number of included studies.

## Data Availability

No new data were created or analyzed in this study. Data sharing is not applicable to this article.

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
