# Peer review of "Integrated Serosurveillance of Infectious Diseases Using Multiplex Bead Assays: A Systematic Review"

_tropicalmed, 2025, doi:10.3390/tropicalmed10010019_

Round 1
Reviewer 1 Report
Comments and Suggestions for Authors
After critically reviewing the provided systematic review, here is an evaluation based on the key sections of a scientific article, focusing on both strengths and areas for improvement:
1).Abstract:
- Strengths: the abstract is concise and outlines the study's objectives, methodology, and findings. It highlights the growing use of multiplex bead assays (MBA) for serosurveillance.
- Weaknesses: the abstract lacks specific numerical results and key statistics to quantify findings, such as percentages or trends. It fails to mention limitations or the scope of the review clearly.The conclusion is vague, making it difficult to understand the impact of the findings.
Suggestion: include specific data points, such as the number of included studies by category and the geographical distribution. A more impactful conclusion is needed to summarize public health implications.
2).Introduction:
- Strengths: provides context on the importance of integrated serosurveillance and the advantages of MBA.
- Weaknesses: repetitive statements make it verbose. There is an overemphasis on generic information about serosurveillance without narrowing down the significance of this review.
Suggestion: streamline the content, reduce repetition, and better define the gaps this review aims to address.
3). Materials and Methods:
- Strengths: adherence to PRISMA guidelines is a strength, and the use of Covidence for data extraction ensures systematic handling of the review process.
- Weaknesses: the inclusion and exclusion criteria are vaguely described, particularly for excluded pathogens or studies. Methods lack sufficient detail about how quality assessments of included studies were conducted.Reference 10 is improperly used as the manuscript itself, which undermines credibility.
Suggestion: xxpand on the criteria for study selection and quality assessment. Address the improper reference and improve transparency in describing the protocol registration.
4).Results:
- Strengths: the results section provides a comprehensive overview of included studies and summarizes the use of MBA across various regions and pathogens.
- Weaknesses: tables are overwhelming (spanning nine pages) and lack synthesis, making it difficult to draw clear conclusions.Trends are described but not critically analyzed.
Suggestion: condense the tables to focus on key data points and highlight patterns or significant findings within the text.
5). Discussion:
- Strengths: provides a thoughtful overview of the implications of MBA in public health and its advantages over traditional methods.
- Weaknesses: the discussion is too general and reiterates results without deeper critical analysis. Insufficient emphasis on limitations of MBA, sampling biases, or practical challenges in implementation.
Suggestion: strengthen critical analysis by discussing potential biases and limitations more rigorously. Include more actionable recommendations for future research.
6).Conclusion:
- Strengths: summarizes the utility of MBA in serosurveillance.
- Weaknesses: the conclusion is repetitive and lacks specificity.There is no call to action or significant takeaway for policymakers or researchers.
Suggestion: include a clear summary of findings with actionable recommendations, and emphasize the potential for MBA to impact global health surveillance.
7). References:
- Strengths: covers a wide range of relevant studies.
- Weaknesses: many references lack DOIs, which compromises accessibility and traceability. The inclusion of the manuscript itself as a reference is highly problematic. Limited inclusion of high-quality, open-access journals undermines the rigor of the review.
Suggestion: replace low-quality references with those from reputable journals and ensure all references have DOIs and are open-access where possible.
8). General Concerns:
- Tables and Figures: while comprehensive, they are excessive and not reader-friendly. Consider summarizing key data in fewer tables and enhancing clarity.
- Writing Quality: the manuscript contains redundancies and lacks precision in several sections. Proofreading for clarity and conciseness is recommended.
- Merit of Rewriting or Rejection: the review demonstrates potential but falls short of being publication-ready due to significant methodological and presentation flaws.If the authors are willing to address these issues thoroughly, a major revision is warranted. Otherwise, rejection might be appropriate due to the improper use of references and the lack of rigor.
9). Final Recommendation: This manuscript requires major revisions to meet the standards of a high-quality systematic review. Authors should focus on: correcting methodological flaws. Improving the clarity and focus of the narrative. Ensuring all references are accurate, traceable, and of high quality.
And please also check my comments in the attached document.

Author Response
Thank you for taking the time to review this paper. We appreciate your careful consideration and the valuable insights you have contributed to improving this study.
Comment 1: Abstract: Strengths: the abstract is concise and outlines the study's objectives, methodology, and findings. It highlights the growing use of multiplex bead assays (MBA) for serosurveillance. Weaknesses: the abstract lacks specific numerical results and key statistics to quantify findings, such as percentages or trends. The conclusion is vague, making it difficult to understand the impact of the findings. Suggestion: include specific data points, such as the number of included studies by category and the geographical distribution. A more impactful conclusion is needed to summarize public health implications.
Response 1: The abstract has been refined to include specific data on the number of studies for each category and a brief overview of the geographic distribution (lines 17-19). A minor error in the percentage calculations was discovered and this was also corrected. Furthermore, the conclusions were strengthened by summarising the potential public health implications of using MBA (lines 27-30). We did not discuss limitations in the abstract due to the word count limit.
Comment 2: Introduction: Strengths: provides context on the importance of integrated serosurveillance and the advantages of MBA. Weaknesses: repetitive statements make it verbose. There is an overemphasis on generic information about serosurveillance without narrowing down the significance of this review. Suggestion: streamline the content, reduce repetition, and better define the gaps this review aims to address.
Response 2: The key gap that is being addressed by this review is an in-depth examination of how MBA has been used for integrated serosurveillance and how this has been implemented and utilised. To the best of our knowledge, no other systematic reviews have addressed this issue. This has been emphasised in the introduction (lines 67-68). In addition, Additionally, we have streamlined the introduction to minimize repetition.
Comment 3: Materials and Methods: Strengths: adherence to PRISMA guidelines is a strength, and the use of Covidence for data extraction ensures systematic handling of the review process. Weaknesses: the inclusion and exclusion criteria are vaguely described, particularly for excluded pathogens or studies. Methods lack sufficient detail about how quality assessments of included studies were conducted. Reference 10 is improperly used as the manuscript itself, which undermines credibility. Suggestion: expand on the criteria for study selection and quality assessment. Address the improper reference and improve transparency in describing the protocol registration.
Response 3: Apologies about the confusion regarding reference #10 – this was supposed to be a reference to the protocol registration through INPLASY. This has been corrected and cited in a more suitable manner (line 80). Details regarding quality assessment and risk of bias (completed using the ROBIS tool) have now been added to the methods (lines 116-122) and can be found in the supplementary material.
Comment 4: Results: Strengths: the results section provides a comprehensive overview of included studies and summarizes the use of MBA across various regions and pathogens. Weaknesses: tables are overwhelming (spanning nine pages) and lack synthesis, making it difficult to draw clear conclusions. Trends are described but not critically analyzed. Suggestion: condense the tables to focus on key data points and highlight patterns or significant findings within the text.
Response 4: was removed as this seemed unnecessary given that the titles for the studies included are in the reference list. Table 2 has now become Table 1 and this has been highly condensed by integrating the information regarding laboratory of analysis and method of blood sample collection into the results section (lines 171-175). The information regarding operational details was also condensed to include only key information on study design and selection of participants within each study. As this review is a narrative description examining how MBA has been used for integrated serosurveillance of infectious diseases, we only provided limited critical analysis of seroprevalence results. However, this has been discussed with a critical lens in the discussion.
Comment 5: Discussion: Strengths: provides a thoughtful overview of the implications of MBA in public health and its advantages over traditional methods. Weaknesses: the discussion is too general and reiterates results without deeper critical analysis. Insufficient emphasis on limitations of MBA, sampling biases, or practical challenges in implementation. Suggestion: strengthen critical analysis by discussing potential biases and limitations more rigorously. Include more actionable recommendations for future research.
Response 5: Repetition of results has been removed from the discussion. The potential bias of sampling methods, a key finding from this review, is now further discussed in the fourth paragraph. This has been amended to include more detailed discussion regarding this potential bias (lines 268-275). Another key limitation discussed in this study is the lack of local capacity to conduct MBA laboratory analysis. This can lead to a delay in timely findings and has been included as a critique in the discussion (lines 310-317). As suggested by the reviewer, actionable recommendations for further research have been added to the discussion and include expanding availability of antigens for inclusion in MBA (lines 317-319), utilisation of findings from integrated serosurveillance studies to be implemented into VPD programs (lines 260-262), approaching serosurveys with sampling designs to target multiple pathogens (lines 280-283) and enabling local capacity for laboratory analysis to occur within countries to streamline the pipeline between research findings and public health actions (lines 313-317).
Comment 6: Conclusion: Strengths: summarizes the utility of MBA in serosurveillance. Weaknesses: the conclusion is repetitive and lacks specificity. There is no call to action or significant takeaway for policymakers or researchers. Suggestion: include a clear summary of findings with actionable recommendations and emphasize the potential for MBA to impact global health surveillance.
Response 6: A more concise and specific summary of the main findings of the review has been added to the conclusion (lines 331-333). Further actionable recommendations regarding MBA utilisation have also been added to the conclusions (lines 334-336), as well as noting the limitations of integrated
Comment 7: References: Strengths: covers a wide range of relevant studies. Weaknesses: many references lack DOIs, which compromises accessibility and traceability. The inclusion of the manuscript itself as a reference is highly problematic. Limited inclusion of high-quality, open-access journals undermines the rigor of the review. Suggestion: replace low-quality references with those from reputable journals and ensure all references have DOIs and are open-access where possible.
Response 7: The reference list has been updated to include DOI for all journal articles and some references of lower quality have been removed. Most of the references used in this review are from high quality open or hybrid access journals such as American Journal of Tropical Medicine and Hygiene (n=18), Malaria Journal (n=9), PLoS NTD (n=11), PLoS One (n=7) and Vaccine (n=9).
Comment 8: General Concerns: Tables and Figures: while comprehensive, they are excessive and not reader-friendly. Consider summarizing key data in fewer tables and enhancing clarity.
Response 8: As discussed in the response to comment 4, Table 2 has now become Table 1 and this has been highly condensed by integrating the information regarding laboratory of analysis and method of blood sample collection into the results section. The information regarding operational details was also condensed to include only key information on study design and selection of participants within each study.
Comment 9: Writing Quality: the manuscript contains redundancies and lacks precision in several sections. Proofreading for clarity and conciseness is recommended.
Response 9: Thank you again for your time in reviewing this study. We have carefully reviewed the manuscript and made the necessary revisions to address the redundancies and improve precision. The sections in question have been proofread for clarity and conciseness, ensuring that the content is now more streamlined and precise.
Reviewer 2 Report
Comments and Suggestions for Authors
Dear Authors,
thank you for your submitted manuscript, providing a comprehensive review of an interesting research issue. Please pay attention to the following questions and comments, pertaining to your manuscript:
1. The use of term “infectious diseases” in the title is very general. Please try to become more specific, like: tropical infectious diseases.
2. Introduction: MBA-based serosurveillance is solely implemented for the estimation of the prevalence and the vaccination coverage against infectious diseases or also of the incidence in a specific population?
3. Literature Search: Data was extracted using Covidence. Title and abstract screening was conducted using Covidence and full text screening was conducted manually?
4. Inclusion Criteria: any meta-analysis included to your search?
5. Figure 1. Please mention in the flow chart why 3450 studies were excluded.
6. Lines 124-126. The summary of the countries, according to their region, is 47 and not 30 as mentioned in the text. Please make this point clear.
7. Line 130. This is Figure number 2 and not 3, as mentioned in the text. Please correct.
8. Lines 165-169. Please try to have an overall of 47 studies, according to their setting. The same please, according to the implemented method, in Lines 169-170.
Best Regards
Author Response
Thank you for taking the time to review this paper. We appreciate your careful consideration and the valuable insights you have contributed to improving this study.
Comment 1: The use of term “infectious diseases” in the title is very general. Please try to become more specific, like: tropical infectious diseases.
Response 1: The term “infectious diseases” was chosen as some of the pathogens included in this review do not fall exclusively under the umbrella of tropical diseases. This includes vaccine-preventable diseases such as tetanus, diphtheria, and pertussis and bacterial pathogens such as Salmonella species. We chose the term “infectious diseases” rather than “tropical diseases and vaccine preventable diseases and enteropathogens” to keep the title concise and inclusive of all pathogens included in this review.
Comment 2: Introduction: MBA-based serosurveillance is solely implemented for the estimation of the prevalence and the vaccination coverage against infectious diseases or also of the incidence in a specific population?
Response 2: To clarify, as serosurveillance is intended to generate evidence of immunity or prevalence of antibodies (rather than clinical cases) at a single point in time. In general, t; however, this may be possible if serosurveillance studies use a repeated survey design such as a cohort or a population-representative longitudinal study design. Further information regarding this has been added to the introduction (lines 63-66).
Comment 3: Literature Search: Data was extracted using Covidence. Title and abstract screening was conducted using Covidence and full text screening was conducted manually?
Response 3: To clarify, the data extraction was also conducted using Convidence. We have revised the paper to clarify this (line 84-85).
Comment 4: Inclusion Criteria: any meta-analysis included to your search?
Response 4: Meta-analysis was not as the focus of this review included multiple different pathogens and sub-analysis of prevalence would be highly limited when grouped into appropriate categories. As for article inclusion within this study, meta-analysis were acceptable within the inclusion criteria, however none were found during article searches. This has been added and addressed in the methods section (lines 118-122).
Comment 5: Figure 1. Please mention in the flow chart why 3450 studies were excluded.
Response 5: These studies were excluded following title and abstract screening and this has been amended in the PRISMA flow diagram.
Comment 6: Lines 124-126. The summary of the countries, according to their region, is 47 and not 30 as mentioned in the text. Please make this point clear.
Response 6: To clarify, 30 is the number of different countries in which the studies were conducted. This does not equate to 47 as multiple studies were conducted in the certain countries.
Comment 7: Line 130. This is Figure number 2 and not 3, as mentioned in the text. Please correct.
Response 7: Thank you for the pickup - This has been corrected.
Comment 8: Lines 165-169. Please try to have an overall of 47 studies, according to their setting. The same please, according to the implemented method, in Lines 169-170.
Response 8: This has been clarified to include the studies where the study setting was not one of these categories (or not defined) (lines 171-173).
Round 2
Reviewer 1 Report
Comments and Suggestions for Authors
Dear authors,
Thank you for responding to my suggestions and understanding that they were intended to improve a manuscript that was already good but needed adjustments before publication. The only error that remains is In version 2 sent, the references continue without the DOI and the authors' names were almost all removed and the format Author1 et al. was used.
Reviewer 2 Report
Comments and Suggestions for Authors
Dear Authors,
thank you for providing comprehensive and convincing answers to the questions and queries expressed by me and the other Reviewers and made changes, that have contributed to the optimization of your manuscript and increased the publishing potential of your work. I have no further questions, pertaining to your manuscript.
Best Regards